# Role of SUMOylation in Neurodegenerative Diseases

**DOI:** 10.3390/cells11213395

**Published:** 2022-10-27

**Authors:** Nicolas Mandel, Nitin Agarwal

**Affiliations:** Institute of Pharmacology, Medical Faculty Heidelberg, Heidelberg University, 69120 Heidelberg, Germany

**Keywords:** post-translational modifications, SUMOylation, neuronal diseases, Alzheimer’s disease, Parkinson’s disease, Huntington’s disease, diabetic peripheral neuropathy

## Abstract

Neurodegenerative diseases (NDDs) are irreversible, progressive diseases with no effective treatment. The hallmark of NDDs is the aggregation of misfolded, modified proteins, which impair neuronal vulnerability and cause brain damage. The loss of synaptic connection and the progressive loss of neurons result in cognitive defects. Several dysregulated proteins and overlapping molecular mechanisms contribute to the pathophysiology of NDDs. Post-translational modifications (PTMs) are essential regulators of protein function, trafficking, and maintaining neuronal hemostasis. The conjugation of a small ubiquitin-like modifier (SUMO) is a reversible, dynamic PTM required for synaptic and cognitive function. The onset and progression of neurodegenerative diseases are associated with aberrant SUMOylation. In this review, we have summarized the role of SUMOylation in regulating critical proteins involved in the onset and progression of several NDDs.

## 1. Introduction

Neurodegenerative diseases (NDDs) impose a substantial medical and financial burden on society worldwide. NDDs are marked by neuronal dysfunction and death, causing progressive degeneration of the structure and function of the central and peripheral nervous system [1,2,3]. A longer life span results in an increased prevalence of age-dependent neurodegenerative diseases. Thus, a significant percentage of the population affected by NDDs is elderly. Inadequate knowledge and a lack of diagnostics tools cause NDDs to have limited options for treatment. Clinical interventions may help to reduce the progression of neurodegenerative diseases, but they fail to cure them permanently. NDDs have a devastating impact on individuals, family and society. The onset of NDDs causes mild symptoms that are easy to ignore.

The progressive loss of a large number of neurons and the miscommunication of neuronal circuits results in memory impairment, learning disabilities, deteriorated synaptic function and cognitive defects [1,4,5,6,7]. NDDs are associated with the transformation and accumulation of misfolded pathogenic proteins [8], mitochondrial dysfunction [9], oxidative stress [10,11], glutamate toxicity and neuroinflammation [12] in neuronal cells, leading to cellular dysfunction and neuronal death. Alteration in post-translational modifications (PTMs) could lead to dysregulation in intracellular processes. This could be because of compromised protein-quality control, altered protein–protein interactions, dysfunctional molecular chaperones, and the impairment of the ubiquitin–proteasome system and lysosomal degradation pathways [13,14]. These cause the accumulation of toxic aggregates and contribute to neuronal degeneration [15]. PTMs are covalent modifications of protein that operate by attaching different chemical entities at single or multiple amino acid residues. These modifications add to the complexity of proteins’ functions by regulating their activity, localization, interactions, life cycle and trafficking [16,17]. The PTMs are classified into several groups: acetylation, phosphorylation, nitration, methylation, ubiquitination and SUMOylation [17,18]. Several studies have demonstrated that aberrant PTM is the reason for pathogenicity associated with NDDs. Herein, we summarize the role of SUMOylation in the onset and progression of neurodegenerative diseases.

## 2. SUMOylation

SUMOylation involves the attachment of small ubiquitin-like modifier (SUMO) proteins to the lysine residues in the protein. Meluh and Koshland first identified it in 1995 [19]. SUMO proteins belong to the ubiquitin-like proteins family (Ubls), characterized by the presence of globular β-grasp fold and a C-terminus Gly-Gly motif in the mature peptide [20,21,22]. In late 1990, SUMO-encoding genes were identified in mammals encoding five different SUMO paralogs, namely, SUMO-1, -2, -3, -4 and -5. SUMO2 and SUMO3 share 97% peptide sequence homology, and differ only in the second and third amino acids and the missing asparagine in SUMO3 [22,23,24,25]. Mature SUMO2 and SUMO3 are 92 and 93 amino acids. The available antibodies cannot differentiate between SUMO2 and SUMO3, collectively referred to as SUMO2/3 [26]. The presence of internal SUMO acceptor lysine K11 in SUMO2/3 enables them to form uniform poly-SUMO chains [23,26]. The additional lysine residue in SUMO2/3 contributes to chain formation under different physiological conditions. SUMO1 is distinct and shares only 47% sequence identity with SUMO2/3. In contrast to SUMO1 and -2/3, SUMO 4 and SUMO5 have not been studied in detail. Guo et al. showed that SUMO4 conjugates to I kappa B alpha (IκB) and negatively regulates NK kappa B (NKκB) transcriptional factor activity [25]. Additional studies have shown the role of SUMO4 in oxidative stress in human placenta tissue [27]. SUMO1 and -2/3 are widely expressed in tissues. In contrast, SUMO4 expression is restricted to the heart, spleen, pancreatic tissues, kidney, brain and lymph node. SUMO5 is highly homologous to SUMO1 [28,29]. It is shown to express in primates’ testes and blood cells, but appears absent in mice. Promyelocytic leukaemia (PML) conjugate to polySUMO5 at lysine 160 residue, which facilitates the formation of promyelocytic leukaemia nuclear bodies (PML-NB) [30] that play a crucial role in regulating transcriptional activity, DNA repair, tumor suppression and apoptosis. Dysregulation in PML-NB generation or function causes polyglutamine repeat-induced neurodegenerative diseases [30]. 

SUMOylation is a dynamic process involving target protein switching between SUMOylated or deSUMOylated states. SUMOylation regulates target protein trafficking and subcellular localization, stability, solubility, activity structural conformations, and protein interactions [31]. In an unstressed cell, only a tiny fraction of the target protein is SUMOylated [32]. Stress or any other physiological signal could change the balance between the SUMO-conjugated and unconjugated protein pools. A SUMOylation burst may alter the modified protein’s solubility and structural conformation. As a result, it could instigate other PTMs, such as phosphorylation, ubiquitination, and acetylation [31]. These modifications can stabilize the substrate protein in the functional complex or regulate the subcellular localization, even after the deSUMOylation event. Alternatively, in response to intrinsic and external stimuli, the SUMOylated fraction of the protein could be trafficked to the target subcellular compartment, or recruited within the functional complex [33]. The emerging concept proposes that SUMOylation modifies multiple proteins within a biological process. SUMOylation may inhibit or activate different components of a biological pathway to integrate the biological process outcome [31,34]. For example, in *Saccharomyces cerevisiae*, multiple proteins involved in DNA repair pathways undergo SUMOylation in response to DNA damage signals. It expedites the DNA repair process [35]. Similarly, in diabetes induced-neuropathy, the SUMOylation of several metabolic enzymes regulates the bioenergetics of cells. This results in the regulation growth of tissue and limits the tissue damage pathways [36]. Although an analysis of individual SUMOylated target proteins is very much needed, new studies deciphering the impact of SUMOylation on different signaling cascades in human disease are a critical approach for the future.

SUMOylation is a reversible post-translational protein modification. Similar to ubiquitination, SUMO moieties form isopeptide bonds with the ε-amino group of acceptor target lysine residue [22,31,37]. SUMOylation is an energy-driven, enzymatic cascade that involves E1-activating enzymes, E2-conjugating enzymes and E3 SUMO ligases (a schematic overview is given in Figure 1). The expressed SUMO forms are C-terminally extended precursors. The SUMO precursors undergo C-terminal cleavage mediated by Sentrin/SUMO-specific protease (SENP) enzymes, which expose the C-terminal diglycine motif of SUMO. The mature SUMO protein binds to the heterodimeric E1 enzyme. The E1 enzyme consists of SUMO-activating enzyme subunits 1 (SAE1) and subunit 2 (SAE2, also known as Uba2). The activation of SUMO is an ATP-dependent two-step reaction. The first step involves the SAE1-mediated adenylation of the C-terminal diglycine motif of SUMO. The structural modulation expedites the binding of adenylated-SUMO molecules via a thioester bond to cysteine residues in the SAE2 enzyme. The SAE2-attached SUMO is then actively transferred to the cysteine residue of the E2-conjugating enzyme. UBE2I (ubiquitin conjugating enzyme E2I) is the human analogue of the E2-conjugating enzyme, and it is also known as UBC9 in yeast [38]. This single known enzyme ensures SUMO binding to the substrate lysine at the consensus site.

In addition to UBC9-mediated target recognition potency, the binding affinity of UBC9 to the target sequence enhances substrate SUMOylation. Additional mechanisms further vary the pool of SUMOylated proteins in a given tissue type. These involve the external or internal stimuli-induced regulation of UBC9 expression. UBC9 activity is modulated via intrinsic post-translation modifications, such as self-SUMOylation at Lys 14 and Cdk1/Cyclin B-mediated phosphorylation [39]. Identifying a single E2-conjugating enzyme is perplexing, and a cause for a pathogen to ambush the SUMOylation pathway to facilitate proliferation in host cells. Mice lacking UBC9 die during embryonic development [40]. The E2-conjugated enzyme promotes the transfer of the SUMO molecule to the target lysine. The transfer generally requires SUMO E3 ligase activity, which stabilizes the interaction between target lysine and SUMO-charged E2-conjugated enzyme. The E3 ligase thereby reinforces the formation of a thioester bond between the target lysine and SUMO molecules [37]. This process does not require interaction between the E3 ligases and target lysine residue in the consensus motif of the substrate protein. SUMO E3 ligases promote quantitative target modification by expediting SUMO conjugation. 

Based on molecular structure, the SUMO–E3 ligases can be classified mainly into the SP-RING domain family, the TRIM superfamily, SIM-containing SUMO E3 ligases and others [41,42,43,44] (list of E3 ligases, Table 1). The SP-RING domain in E3 ligases binds to UBC9 and stimulates SUMO ligation to target lysine residues. Protein inhibitors of activation STAT (PIAS) proteins were the first identified member of the SP-RING domain family of enzymes [45]. Furthermore, PIAS proteins contain a subsidiary SUMO-interaction motif (SIM) that promotes SUMO-UBC9-PIAS complex formation, increasing SUMOylation efficacy on the target lysine residues [43]. The TRIM superfamily of E3 ligases has approximately 100 members. They share the common tripartite motif (TRIM) consisting of the RING domain, B-box domains and coiled-coil region [41]. The RING domain commences UBC9 binding, whereas the B-box domain binds to the substrate, which enhances the SUMO transfer from UBC9 to target lysine [44]. SIM-containing E3 ligases form the third type of ligases in the SUMOylation pathway, where SIMs are the short stretch of hydrophobic residues with an acidic or polar residue at position 2 or 3. The SIM motif sequence forms a tetra core sequence of ψ X ψ ψ or ψ ψ X ψ. In the consensus sequence, “ψ” is the hydrophobic amino acid with a large non-polar side chain, typically Isoleucine (I), Valine (V) or Leucine (L), and X is any amino acid. The hydrophobic core is flanked on one side or the other by negative charged Aspartic acid (Asp), Glutamic acid (Glu) or Phosphoserine (Ser) residues. Hydrophobic sequences facilitate SUMO interaction. RanBP2, CBX4, SLX4 and ZNF451 are some of the more well-studied E3 ligases in this family [46,47,48,49]. Even though several E3 ligases have been identified, further investigation is required to understand their functionality in substrate structure–function recognition. SENPs can rapidly remove the SUMO attached to a lysine residue. This process is known as deSUMOylation.

Proteomic analysis revealed that most acceptor lysine residue is in the consensus motif. The classical consensus SUMOylation motif is ψKXE/D, where ψ represents hydrophobic residue, K is the acceptor lysine residue, X represents any amino acid, and E/D is glutamic or aspartic acid residues, respectively [75]. ψ contributes to the binding of UBC9, and its mutation abolishes complex formation and the SUMOylation of the target lysine. In addition, several other SUMOylation motifs were identified. They include the inverse consensus motif E/DXKψ [76], phosphorylation-dependent SUMOylation motif (PDSM; ψKXEXX(pS)P), where P is proline residue [77], hydrophobic consensus motif (ψψψKXE) [76] and negatively charged amino acid-dependent SUMOylation motif (NDSM; ψ KXEXXEEEE) [78]. These PDSMs or NDSMs represent an additional way to regulate the substrate SUMOylation in an inducible manner. The proximity of the phosphorylation site/negatively charged motif to the SUMOylation motif directly affects the substrate–UBC9 affinity and thus introduces a further regulatory steps [77,78]. Table 2 summarizes the different SUMOylation motifs. 

Several studies have established the role of SUMOylation in the onset and progression of NDDs. Aberrant SUMOylation causes the aggregation of the toxic proteins, defects in protein trafficking, the modification of the ion channel properties and the release of neurotransmitters in neurons, oxidative stress and alteration in gene expression. Altered SUMOylation potentially modifies the functions of hundreds of proteins in many different pathways, and results in cognitive deficits associated with NDDs.

## 3. Role of SUMOylation in Alzheimer’s Disease (AD)

AD is the dominant form of escalating dementia. AD primarily occurs in an ageing population, but hereditary or non-hereditary mutations, lifestyle and environment may contribute to the onset and progression of AD pathogenicity. Currently, there are more than 50 million AD patients, and by 2050 these numbers are predicted to reach 152 million [6]. Clinically, AD is diagnosed and marked by initial memory dysfunction and, in different advanced stages, is associated with multiple cognitive dysfunctions. AD is associated with the aggregation of extracellular amyloid plaques and intracellular neurofibrillary tangles (NFT), gliosis, and the extensive loss of neurons and synapses. Amyloid plaques are primarily composed of deposits of beta-amyloid protein (Aβ) formed by the proteolytic cleavage of the transmembrane amyloid precursor protein (APP) by beta-secretase (β-secretase) and gamma-secretase (γ-secretase). In the hippocampus, cortex and amygdala, aggregating these plaques engenders neurotoxicity, causes astrocytes and microglia stimulation, and results in neuronal dysfunction and degeneration [79,80,81,82]. NFTs are abnormally paired helical filaments of the hyperphosphorylated Tau proteins. Under physiological conditions, Tau protein binds to microtubules and assists in their stabilization and neuronal transport. However, the hyperphosphorylation of Tau causes microtubules disintegration. It results in the accumulation of NFT in the neuralperikaryal, cytoplasm, axons and dendrites, which causes neuronal devastation and degeneration. This highly complex interplay results in oxidative and mitochondrial stress that causes synaptic homeostasis loss and neuronal loss [83,84].

Cell culture-based in vitro systems and a transgenic mouse model of AD were used to study the impact of SUMOylation on AD pathology. Expression studies in the early-aged Tg2576 mouse model of AD revealed no changes in the global levels of SUMO-1 [85]. In contrast, SUMO-2/3 showed several disparities in their expression patterns. However, aged Tg2576 mice exhibit excessive levels of SUMO-1 conjugated proteins in cortical and hippocampal tissue compared to age-matched wildtype mice [85]. Electrophysiological studies in hippocampal neurons manifested that impaired SUMOylation contributes to deficits in learning and memory pathology associated with Alzheimer’s disease [86]. The overexpression of UBC9 rescued Aβ-mediated deficits in long-term potentiation (LTP) and hippocampal-dependent learning and memory in a mouse model of AD [86]. UBC9 is highly mobile in neurons [87]. The Aβ peptide facilitates the redistribution of the UBC9 protein from presynaptic to postsynaptic terminals. Extracellular vesicles (EV) are responsible for the release of UBC9 in the presynaptic neuron, and also may facilitate the synaptic gap transmission. The presence of the Aβ peptide results in Synaptosome Associated Protein 23 (SNAP-23) phosphorylation, facilitating the release of UBC9 from EVs, promoting the development of AD [88]. In addition, APP is SUMOylated at lysine 587 and 595 [89]. Proximity to the β-secretase cleavage site raises the possibility of the SUMOylation-mediated modulation of Aβ levels. Site-directed mutagenesis studies of APP SUMOylation sites evince that the deSUMOylation of APP promotes Aβ levels. Over-expressing genes encoding enzymes of SUMOylation pathways ensure raised APP SUMOylation and decreased Aβ aggregates [89,90]. In contrast, SUMOylation positively modulates the expression of APP-cleaving protease β-secretase-encoding gene [89]. This results in increased levels of Aβ aggregates [91]. These paradoxical findings enhance the understating of SUMOylation-mediated pathological changes in AD. A recently published in vitro study on cultured cortical neurons demonstrated that Aβ_1-42_ peptide downregulates the levels of SUMO-activating enzyme SAE2 and the SUMO E3 ligase PIAS1/2. This results in decreased levels of SUMO1 or SUMO2/3-conjugated proteins [92]. Moreover, the impact of SUMOylation on APP trafficking remains unexplored. The presence of phosphorylated Tau protein is associated with AD pathology. Biochemical studies revealed that the Tau protein lysine K340 shared the SUMOylation and ubiquitination modifications. SUMOylation at K340 blocks ubiquitin binding to Tau and, thereby, its proteasomal degradation [93]. The activity-based signal determines the balance between SUMOylation and ubiquitination, and thus, the fate of the Tau protein. The histopathology of AD mice revealed that SUMO1 colocalized with phosphorylated Tau aggregates [93]. Furthermore, increased SUMO1 immunoreactivity was detected in Tau aggregates in AD patients’ cerebral cortex [94]. Pharmacologically inhibiting proteasomal degradation manifests Tau SUMOylation, causing Tau phosphorylation and tangle formation. The results demonstrate that the SUMOylation of Tau protein endorses Tau hyperphosphorylation. It leads to reduced microtubule stability and neuronal dysfunction [94]. This is the preliminary evidence depicting the critical role of SUMOylation in the onset and progression of AD in humans and rodents. There are still several open questions concerning the role of SUMOylation in glia and astrocytes, which are known to modulate neuronal activity and can contribute to the progression of AD. Few studies indicate that SUMOylation is involved in redox imbalance in astrocytes. This redox imbalance may cause neuroinflammation and pathophysiological symptoms associated with AD [95]. The role of SUMOylation in the glia-mediated modulation of neuronal dysfunction is an emerging concept and needs further investigation. Figure 2 shows an overview of the different mechanisms by which SUMOylation plays a role in AD.

## 4. Role of SUMOylation in Parkinson’s Disease (PD)

PD is a multifactorial neurodegenerative disease. It is manifested by the progressive loss of dopaminergic neurons projecting from the substantia nigra compacta to the striatum. PD-associated neurological disorders include motor and nonmotor symptoms. The early pathological symptoms are rapid eye movement, sleep behavioral disorder and decreased smell. The progression of the disease is associated with the onset of cognitive impairment and hallucinations. The hallmark of PD progression is motor dysfunction. It includes tremors, rigidity, bradykinesia, gait stiffness, hypokinesia and imbalance [96,97,98]. PD prevalence is increasing rapidly. In 2016 there were 6.1 million PD patients globally. Currently, this number has increased to more than 10 million. The incidence of PD increases with age, peaking between 85 and 89 years. However, 4% of people with PD are diagnosed before 50. PD is 1.4 times more prevalent in males than females [99,100]. 

PD is pathological and primarily marked by the cytoplasmic deposition of Lewy bodies, a neuronal inclusion consisting of α-synuclein protein aggregations. The intracellular aggregation of α-synuclein causes decreased membrane potential. Furthermore, it results in oxidative stress, excitotoxicity, neuroinflammation and finally, apoptosis of the neuron [96]. The sequence of molecular alterations is not well delineated. However, it may lead to dysregulation of DJ-1 (protein deglycase DJ-1, encoded by *PARK7* gene) transcriptional factors, an impaired ubiquitin-proteasome system, aberrant splicing events, mitochondrial dysfunction and neuronal oxidative stress. These could potentially lead to the deposition of Lewy bodies in neurons. Lewy bodies are the eosinophilic cytoplasmic inclusions of 5-30 µm spherical masses with a dense core surrounded by radiating fibrils. Lewy bodies are strongly stained for α-synuclein protein [101]. In addition, Lewy bodies consist of neurofilament protein, ubiquitin-binding protein p62, tubulin and synphilin-1. The abundance of α-synuclein in Lewy bodies establishes its importance in the pathogenesis of PD. α-Synuclein is a 140 amino acid protein abundant in presynaptic nerve terminals. In a monomeric state, it is involved in the vesicular transport and storage, release and recycling of neurotransmitters [102]. *SCNA* gene encodes the α-synuclein protein. Autosomal dominant missense mutations in the *SNCA* gene lead to the early onset of PD [103]. In addition, *SCNA* gene multiplication (duplication and triplication of SCNA gene) increases the dose of wildtype proteins and can cause autosomal dominant parkinsonism. A significant number of genetic, transgenic and viral-based studies in rodents and primates, using either wildtype or mutant α-synuclein protein, clearly demonstrate an α-synuclein association with the progressive degeneration of neurons, and thus its crucial role in the pathogenesis of PD. α-Synuclein oligomerization promotes its binding to the mitochondrial outer membrane translocase (TOM20) and the voltage-dependent anion channel (VDAC). It results in the inhibition of the mitochondrial import system [104]. This eventually leads to disrupted mitochondrial–ER interaction, decreased mitochondrial ATP production, mitochondrial swelling and depolarization, consequently causing oxidative stress and accelerated cytochrome C release in neurons and, lastly, neuronal apoptosis [105]. α-Synuclein is colocalized with ubiquitin and 20S proteasome in Lewy bodies. The direct interaction of α-synuclein and 20S proteasome causes the inhibition of the chymotrypsin-like proteasome and a dysfunctional ubiquitin–proteasome system (UPS) [106]. Compromised UPS results in the accumulation of misfolded proteins and builds toxic aggregates [107,108]. NDDs begin with the alteration at synaptic transmission. Wildtype α-synuclein promotes SNARE-complex assembly. SNARE (soluble *N*-ethylmaleimide-sensitive factor attachment protein receptor) mediates membrane fusion at synaptic terminals. SNARE proteins are present on vesicles (v-SNARE) or target membranes (t-SNARE). The C-terminus of α-synuclein binds to VAMP2 (vesicle-associated membrane protein 2) and promotes SNARE complex formation. α-synuclein overexpression or aggregates inhibit v-SNARE binding to t-SNARE, which impairs neurotransmitter release and thus leads to synaptic dysfunction [109,110]. α-synuclein binds to microtubules’ tubulin α2β2 tetramer. It stabilizes the α-synuclein helical structure, which promotes microtubule nucleation and growth. Mutant α-synuclein does not undergo tubulin-induced structural change. It results in tubulin aggregation, rather than polymerization [111,112]. Inhibited axonal transport and synaptic dysfunction further drivie neuronal degeneration [113,114]. 

α-Synuclein post-translational modification (PTM) may influence its toxicity and aggregation. Phosphorylation, nitration, acetylation, ubiquitination, O-GlcNAcylation and SUMOylation extensively modify α-synuclein [115,116,117,118]. The PTM of α-synuclein modifies its membrane binding, protein interaction, solubility and aggregation properties. PD pathogenicity is linked to α-synuclein phosphorylation at serine 87 and 129 [116,119]. In contrast, SUMOylation prevents α-synuclein toxicity. α-Synuclein is SUMOylated at multiple lysine residues. Among them, lysine 96 and 102 were shown to be critical in regulating α-synuclein toxicity [85,120]. Substitutions of lysine 96 and 102 to arginine impaired α-synuclein SUMOylation and reduced its solubility. This results in enhanced aggregation and toxicity in HEK cells and the substantia nigra neurons in a rat PD model [85]. Similar observations were made in transgenic mice expressing His-tagged SUMO2 [85]. SUMOylation preferentially and effectively occurs on α-synuclein lysine102 residue. In vitro studies have demonstrated that the SUMOylation of only 10% of α-synuclein delayed its aggregation [85]. α-Synuclein contains two SIM motifs located between the amino acids 33 and 58. The synthetic SUMO1-derived fragment SUMO1(15-55) can bind to both SIMs, and it prefers SIM-binding defective mutants that only possess one SIM. SUMO1(15-55) inhibits α-synuclein oligomerization by modulating the monomeric state and preventing aggregation in SH-SY5Y neuronal cells. It was also shown that the expression of SUMO1(15-55) in a transgenic α-synuclein *Drosophila* model exerts neuroprotective effects [121]. Lewy bodies are also associated with the parkin protein (encoded by the *PARK2* gene) [122]. Mutations in *PARK2* are the cause of autosomal recessive juvenile parkinsonism [123]. Parkin is a ubiquitin E3 ligase that regulates UPS and lysosomal degradation pathways. It plays a crucial role in degrading misfolded proteins, clearing reactive oxygen species, mitophagy, and promoting cellular survival [124]. Parkin directly and selectively non-covalently interacts with SUMO1 in vitro and in vivo. The SUMOylation of parkin promotes its nuclear transport and self-ubiquitination, which causes a loss of its E3 ligase activity and decreases levels in the cytoplasm. In contrast, SUMOylation promotes parkin transcriptional factor function, resulting in the expression of neuroprotective genes [125]. Neurons deprived of parkin protein are susceptible to oxidative stress, mitochondrial dysfunction and impaired proteasome-mediated protein degradation. Thus, parkin is critical for cellular function and viability [126]. In addition to parkin, the DJ-1 (protein deglycase DJ-1, encoded by *PARK7* gene) protein contributes to detecting and modulating neuronal oxidative stress. It functions as a regulator of mitochondrial biogenesis and a stress regulator. Mutations in DJ-1 are identified in PD patients [127,128]. The Leucine166 Proline substitution causes aberrant SUMOylation in vitro, and in vivo studies identified that DJ-1 is SUMOylated at lysine 130 residue [129]. The SUMOylation of DJ-1 promotes its activity, and thus its ability to affect oxidative stress, mitochondrial function and cell survival [129,130]. 

It could be hypothesized that the SUMOylation of multiple key proteins involved in mitochondrial biogenesis reinforces cell survival, whereas deSUMOylation causes PD pathogenesis. Figure 3 depicts how SUMOylation can affect Parkinson’s disease. 

## 5. Role of SUMOylation in Huntington’s Disease (HD)

Huntington’s disease is a rare, incurable, neurodegenerative, monogenetic, dominantly inherited disease. It is a polyglutamine (poly Q) disorder of Huntingtin protein (HTT). It is caused by the expansion of a CAG repeat (encoding glutamine) at the exon 1 of the Huntingtin (*HTT*) gene [131,132,133]. The onset and progression of HD are associated with the presence of more than 36 CAG triplet repeats encoding polyQ tract in the HTT protein. Patient data showed that 40–50 repeats cause the onset of disease in adults, whereas 50–120 repeats lead to juvenile HD [133,134,135,136]. The *HTT* gene encodes the 348 kDa Huntingtin (HTT) protein found in the cytoplasm and nucleus of neurons. The HTT protein plays a critical role in microtubule-based axon trafficking, which involves nuclear–cytoplasmic protein trafficking and antero- and retrograde axonal transport [137,138]. The HTT competence of bidirectional switching between antero/retrograde axonal transport is phosphorylation-dependent. The phosphorylation of Serine 421 residue of the Huntingtin protein promotes anterograde transport, whereas nonphosphorylated Huntingtin is likely to undergo retrograde transport [139]. In addition, HTT mitigates mitochondria transport along neurites [140]. Homozygous *HTT* mice are embryonically lethal [141,142]. Conditional *HTT* knockout mice showed impaired vesicular and mitochondrial trafficking in both directions, and progressive brain degeneration [143]. The *HTT* gene with the abnormally high numbers of CAG triplet repeats (polyQ) is termed mutant HTT (mHTT). mHTT mitigates the HD disease pathology mainly in striatal medium-sized spiny neurons, cortical pyramidal neurons and thalamic neurons. However, mHTT is susceptible to aggregation, which causes neuronal dysfunction and death [144,145]. HD is an incurable neurodegenerative disease that usually appears in a subject’s 40s. HD is associated with motor, cognitive and psychiatric disorders [146,147,148,149]. 

The motor symptoms include involuntary movements, impaired voluntary movements, speech difficulties and unusual eye movements, whereas cognitive symptoms include learning deficit and lack of impulse control. The neuropsychiatric symptoms include depression, insomnia, frequent fatigue and suicidal tendencies. Several post-translational modifications modify Huntingtin’s protein. Ubiquitination and SUMOylation antagonistically modify the same lysine residue [108,150,151,152,153]. The ubiquitination of HTT causes its proteasome-mediated degradation, whereas the SUMOylation of the identical lysine residues protects it from degradation. SUMOylated HTT may be responsible for toxic accumulation. In contrast, the SUMOylation of the protein fragment encoded by exon 1 of the *HTT* gene (HTTex1) has a paradoxical effect. It leads to a reduced ability to form aggregates and increased solubility, and promotes its ability to repress transcription [154]. In addition, the SUMOylation of HTTex1 inhibits its interaction with lipids mediated by the lipid-binding domain to prevent aggregation [155]. SUMO E3 ligase PIAS1 mediates the SUMOylation of HTT. The downregulation of PIAS1 in the striatum reduces HTT SUMOylation and improves motor impairment independent of mutant HTT presence [156,157]. PIAS1, through interaction with HTT, is also part of the transcription-coupled repair complex (TCR) and polynucleotide kinase 3′-phosphatase (PNKP). PIAS1 can modulate the activity of PNKP in the mouse brain as well as in neurons derived from HD patients [156]. Rhes (Ras-homolog enriched in the striatum) is a multifunction protein. In striatal neurons, Rhes acts as a SUMO E3 ligase, and mitigates the SUMOylation of the mHTT protein and associated cytotoxicity [66,153]. Rhes can transit from neuron to neuron and facilitate the spreading of mHTT [158]. Rhes knockout mice are neuroprotective, indicating that Rhes-mediated SUMOylation is a critical event in mHTT-driven neuronal pathology [159,160,161,162]. mHtt also impairs mitochondrial function [163,164]. A rodent model of HD and human brains affected by HD demonstrated aggregated mHTT proteins in mitochondria [159,165]. Biochemical experiments provided evidence that mHTT interacts with the mitochondrial translocase of inner membrane 23 (TIM23), and mitochondrial fission proteins dynamin-related protein 1 (DRP1) and fission 1 (FIS1) [165,166,167,168,169]. mHTT–TIM23 interaction resulted in a mitochondrial protein import defect in in vitro and in vivo studies [165,168,170]. Moreover, mHTT’s interaction with DRP1/FIS1 promotes mitochondrial fission via the increased expression of mitochondrial fission genes DRP1 and FIS1. In addition, mHTT–DRP1 interaction is also associated with the increased enzymatic activity of GTPase DRP1 [166].

Collectively, mHTT directly contributes to mitochondrial dysfunction and neuronal death. However, SUMOylation has a dual effect on mHTT-mediated effects. Depending upon the SUMO paralog attached to mHTT, it could either improve the cellular clearance of mHTT or enhance its toxicity by stabilizing the aggregates. Thus, SUMOylation plays a critical role in the pathophysiology of HD. A summary of the role of SUMOylation can be seen in Figure 4.

## 6. Role of SUMOylation in Diabetic Peripheral Neuropathy (DPN)

Diabetic peripheral neuropathy is a chronic complication caused by prolonged hyperglycemia in type 1 and 2 diabetes. It is a slowly progressing, inevitable diabetic complication. More than 50% of patients who have diabetes develop neuropathy, and it is diagnosed late [171,172]. Clinically, DPN is characterized by lower limb complications such as foot ulcers, paresthesia, numbness and neuropathic pain [173]. Depending upon the affected neuron type, it could be sensory, motor or autonomic neuropathy. In addition, DPN leads to axonal degeneration, a reduction in intraepidermal nerve fibers, diminished conduction velocities, and demyelination [36,174,175,176,177]. These symptoms may be sporadic or constant, and can be associated with painful or painless DPN.

Typically, glucose is metabolized through glycolysis to pyruvate. Pyruvate then undergoes oxidative decarboxylation to produce acetyl coenzyme in mitochondria. Acetyl coenzyme is further oxidized to carbon dioxide and water through the tricarboxylic acid cycle. The glucose breakdown yields 4 ATP and 12 molecules of NADH. The generated NADH combines with oxygen by passing electrons through the electron transport chain to form 34 ATP molecules [178]. Under hyperglycemic conditions, the excess glucose is shuttled to the toxic aldose reductase polyol pathway, protein kinase C activation, and generates toxic secondary metabolites and advanced glycation end products (AGEs) [36,179,180,181]. The PTMs, such as ubiquitination, phosphorylation, acetylation, and ribosylation, were shown to regulate the glycolytic enzymes involved in complex glucose metabolism [182]. For example, ubiquitination inhibits biphosphate aldolase (ALDOA) and glyceraldehyde-3-phosphate dehydrogenase (GAPDH) levels and activity. The deubiquitination of glycolytic enzymes stabilizes them and promotes glycolysis [183]. Similarly, the acetylation of lysine residues of the glycolytic enzyme blocks substrate binding and impairs metabolic pathways [184]. ]. In contrast, the role of SUMOylation is not well-defined. Only a few studies have attempted to delineate the role of SUMOylation in DPN. The LC-MS-based screening in dorsal root ganglia (DRG) neurons identified several critical enzymes in the glycolysis and Krebs cycle as targets for SUMOylation [36]. The activities of GAPH, citrate synthase, transketolase-like protein-1 (TKTL1) and isocitrate dehydrogenase 2 (IDH2) were shown to be modulated by SUMOylation [185]. In addition, detoxifying enzymes such as aldehyde dehydrogenase isoform α-3A1 (ALDH3A1) are SUMOylation targets [36]. The human sciatic nerve analysis showed that prolonged diabetes causes reduced levels of SUMOylated GAPDH and citrate synthase enzymes involved in glucose metabolism. These findings illustrate the importance of SUMOylation as a neuron-protective mechanism in DPN. Similarly, Hou et al. depicted the role of SUMOE3 ligases in the development of DPN. PIAS1 mediates Peroxisome Proliferator-Activated Receptor γ (PPAR-g) SUMOylation, and stabilizes it. This results in miR-124 upregulation, which inhibits STAT3 (Signal transducer and activator of transcription 3) activity and thus inhibits Schwann cells’ apoptosis and the progression of DPN [186]. In type 1 or 2 diabetes, prolonged hyperglycemia downregulates SUMOylation in neurons and Schwann cells [186]. These findings indicate that the SUMOylation of multiple proteins in several metabolic pathways is essential for endogenous protection from metabolic damage. Tissue-specific genetic ablation studies in rodents deleting the *UBC9* gene abolish the SUMOylation of proteins in dorsal root ganglia neurons. These neurons were shown to be more susceptible to hyperglycemia-induced damage. Thus, these mutant mice demonstrate expeditious peripheral neuropathic pathological symptoms. These findings demonstrate that SUMOylation is a critical protective mechanism, the loss of which promotes hyperglycemia-induced tissue damage. 

In addition, different studies have identified several ion channels, such as sodium channels 1.7 and 1.2 (Nav1.7 and Nav1.2), Transient receptor potential V1 (TRPV1), two-pore potassium channel (K2P1), and multiple voltage-gated potassium channels (Kv1.5, Kv2.1, Kv4.2, Kv7.1, Kv7.2 and Kv11.1), as SUMOylation targets [187,188,189,190,191]. Among these, TRP channels are the most studied ion channels under the pathological condition of DPN. TRPV1 is a cation channel and acts as a thermosensor. It is known to be activated by capsaicin, low pH and noxious heat > 42 °C [192]. In animal models for diabetic neuropathy, TRPV1 levels and function reduction were reported, and linked to the DPN pathophysiology [193]. Recently published studies reported TRPV1 SUMOylation at lysine 324 [36]. Electrophysiological recording and calcium imaging data elucidate the role of SUMOylation in TRPV1 channel function modulation in in vitro and in vivo conditions. In diabetic patient nerve samples, the levels of sumoylated-TRPV1 were shown to be diminished. En masse, SUMOylation increases neuronal excitability, but the differential regulation of individual ion channels under diabetic conditions has not yet been studied.

Prolonged chronic hyperglycemia is associated with excessive cellular oxidative stress and impaired mitochondrial function [194]. In vitro studies on cultured DRG neurons exposed to elevated levels of glucose exhibit increased levels of reactive oxygen free radicals (ROS) and hydrogen peroxide (H_2_O_2_) [174]. ROS/H_2_O_2_ attack the iron–sulfur centers of enzymes involved in glucose metabolism, such as complex I–III in mitochondria, and consequently inhibit their function [195]. This leads to bioenergetic crises, decreased biological activity, and impaired cellular signaling and transport. All these together cause neuronal dysfunction and neurodegeneration. SUMOylation and oxidative stress are bidirectional regulatory mechanisms. SUMOylation activates a signal cascade and transcriptional factors to increase detoxifying protein levels, inhibiting oxidative stress. In beta-pancreatic tissue, oxidative stress activates transcription factor NF-κB via the increased phosphorylation and subsequent degradation of IkB. Similarly, oxidative stress activates the JNK/c-Jun and Maf/Nrf2 pathways in different tissue types [196]. Prolonged chronic hyperglycemia generates H_2_O_2_, which transiently inactivates SUMO E1 and E2 enzymes. It induces the formation of a disulfide bond between their catalytic cysteines [197], which may lead to reduced levels of SUMOylated proteins in the neurons. It is hypothesized that enhanced oxidative stress impaired the SUMOylation-mediated cellular protective mechanism, which could lead to peripheral neuron dysfunction and degeneration. Although a more detailed analysis is required at different levels of disease progression, as yet unknown regulation mechanisms via SUMOylation should be explored as new drug targets to impede the progression of DPN. Figure 5 shows how SUMOylation can modulate DPN.

## Figures and Tables

**Figure 1 cells-11-03395-f001:**
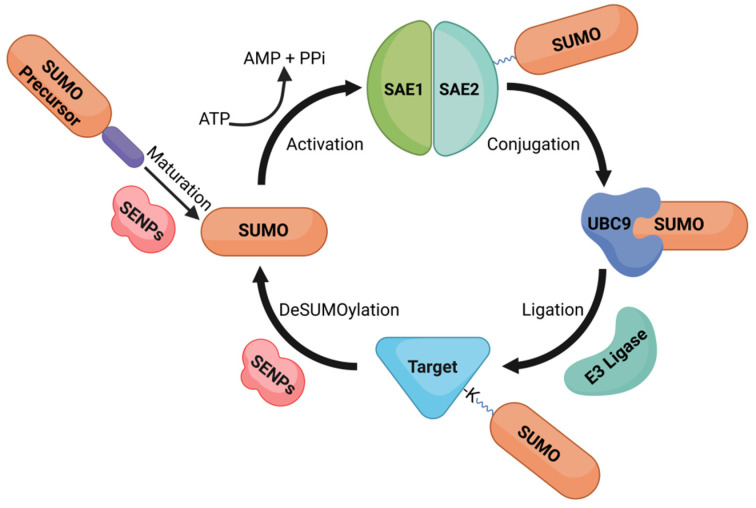
The catalytic cycle of SUMOylation and deSUMOylation. Maturation: SENPs process the SUMO precursor into mature SUMO. Activation: SUMO is activated in an ATP-dependent 2-step reaction by the heterodimeric E1 enzyme and bound via a thioester bond on cysteine residues to the SAE2 subunit. Conjugation: SUMO is actively transferred to the cysteine residue of the E2 enzyme UBC9. Ligation: SUMO is attached to a specific lysine residue in the substrate; this step can be directly done by E2-SUMO but usually requires an E3 ligase that enhances the conjugation. DeSUMOylation: SUMO proteins are cleaved off from substrates by SENPs, free SUMO is available for another catalytic cycle. Figure was created with BioRender.com.

**Figure 2 cells-11-03395-f002:**
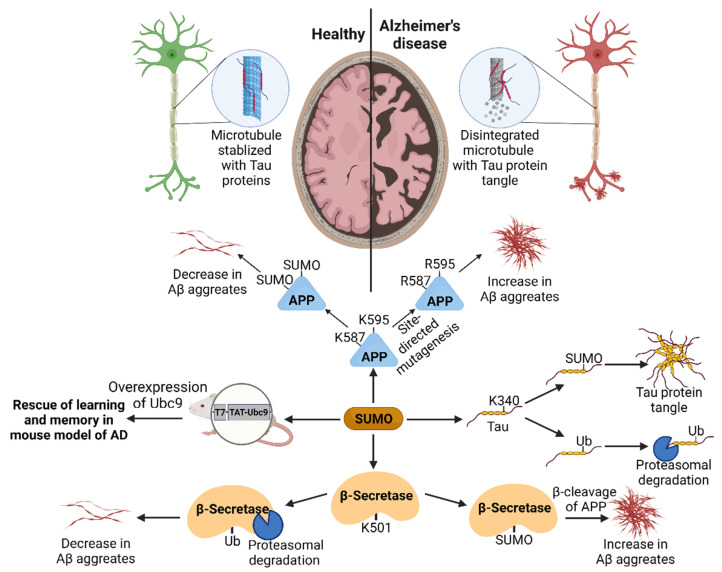
SUMOylation in Alzheimer’s disease: SUMOylation of APP leads to lower levels of Aβ aggregate levels. In contrast, the SUMOylation of β-secretase leads to the increased formation of Aβ aggregates. The SUMOylation of Tau prevents ubiquitination and leads to the formation of Tau protein tangles. UBC9 overexpression was able to rescue Aβ-mediated deficits. Figure was created with BioRender.com.

**Figure 3 cells-11-03395-f003:**
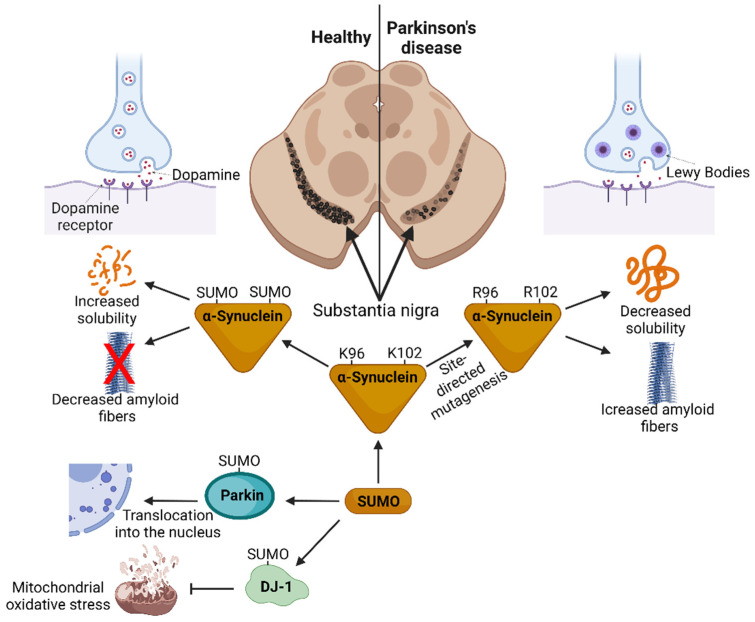
SUMOylation in Parkinson’s disease: SUMO-modified α-synuclein remains soluble and less amyloid fibers are formed. SUMOylation of DJ-1 promotes its activity and results in reduced mitochondrial stress. The SUMOylation of parkin promotes its nuclear transport. Figure was created with BioRender.com.

**Figure 4 cells-11-03395-f004:**
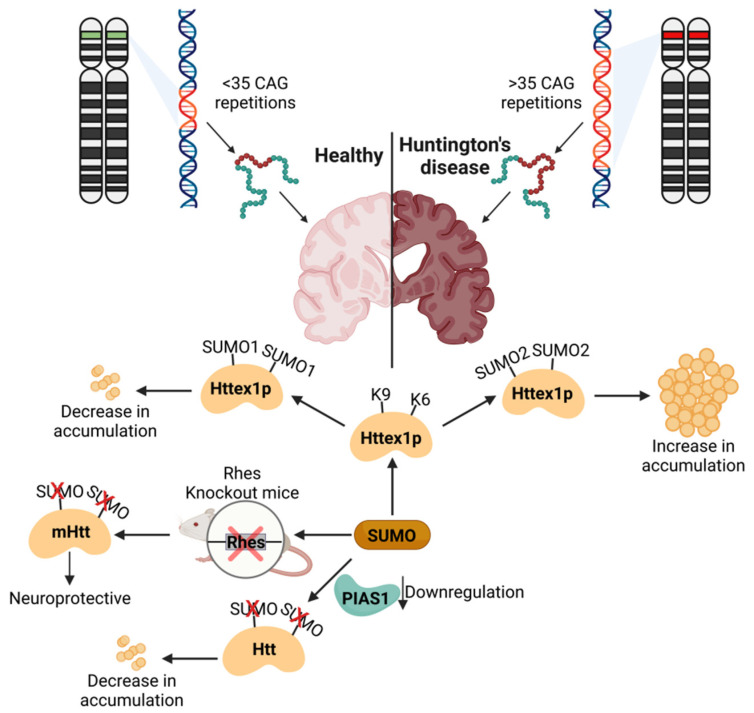
SUMOylation in Huntington’s disease: SUMOylation of Httex1p has an ambivalent effect and can either decrease or increase the solubility, depending on if it is SUMO1- or SUMO2-attached. The downregulation of PIAS1 in the striatum leads to the reduced SUMOylation of HTT and decreased accumulation. Rhes knockout in mice has been shown to have beneficial and neuroprotective effects in Huntington’s disease. Figure was created with BioRender.com.

**Figure 5 cells-11-03395-f005:**
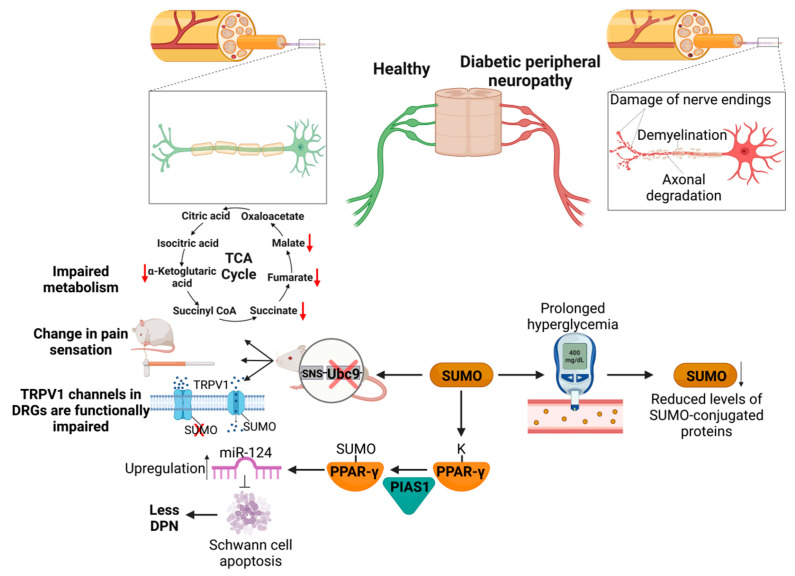
SUMOylation in diabetic peripheral neuropathy: Prolonged hyperglycemia leads to reduced levels of SUMO-conjugated proteins in neuron and Schwann cells. The tissue-specific knockout of SUMOylation in peripheral sensory neurons of DRGs leads to a functional impairment of TRPV1 channels, altered the function of metabolic enzymes and increased the onset of DPN. The SUMOylation of PPAR- γ results in higher miR-124 levels, which inhibit Schwann cell apoptosis. Figure was created with Biorender.com.

**Table 1 cells-11-03395-t001:** Categorization of SUMO E3 ligases.

**SP-RING Family**	**Described by**	**SIM Family**	**Described by**
PIAS1	Kahyo et al. 2001 [45]	CBX4	Kagey et al. 2003 [46]
PIAS3	Sentis et al. 2005 [50]	KIAA1586	Eisenhardt et al. 2015 [48]
PIAS3-β	Sentis et al. 2005 [50]	RanBP2	Pichler et al. 2002 [47]
PIASx-α	Kotaja et al. 2002 [51]	SLX4	Guervilly et al. 2015 [49]
PIASx-β	Acosta et al. 2005 [52]	ZNF451	Eisenhardt et al. 2015 [48]
PIASy	Wong et al. 2004 [53]		
PIASy-E6	Wong et al. 2004 [53]		
MMS21	Potts and Yu 2005 [54]		
ZMIZ1	Moreno-Ayala et al. 2015 [55]		
ZMIZ2	Moreno-Ayala et al. 2015 [55]		
**TRIM Superfamily**	**Described by**	**Others**	**Described by**
TRIM1	Chu and Yang 2011 [41]	ARF	Tago et al. 2005 [56]
TRIM9	Chu and Yang 2011 [41]	BCA2	Lluch and Moreno 2017 [57]
TRIM11	Zhu et al. 2020 [58]	DREF	Yamashita et al. 2016 [59]
TRIM19	Chu and Yang 2011 [41]	HDAC4	Zhao et al. 2005 [60]
TRIM22	Chu and Yang 2011 [41]	HDAC-5, -7, -9	Grégoire and Yang 2005 [61]
TRIM27	Chu and Yang 2011 [41]	KROX2	Gutiérrez et al. 2011 [62]
TRIM28	Chu and Yang 2011 [41]	MAPL	Braschi et al. 2009 [63]
TRIM32	Chu and Yang 2011 [41]	MDM2	Stindt et al. 2011s [64]
TRIM33	Ikeuchi et al. 2014 [65]	RHES	Subramaniam et al. 2009 [66]
TRIM36	Chu and Yang 2011 [41]	RNF	Qiao et al. 2014 [67]
TRIM38	Hu et al. 2017 [68]	RSUME	Nagashima et al. 2007 [69]
TRIM39	Chu and Yang 2011 [41]	SF2	Pelisch et al. 2010 [70]
TRIML2	Kung et al. 2015 [71]	TOPORS	Weger et al. 2003 [72]
		TRAF7	Morita et al. 2005 [73]
		UHRF2	Oh and Chung 2013 [74]

**Table 2 cells-11-03395-t002:** The different SUMOylation motifs.

SUMOylation Motifs	Amino Acids	Described by
Classical Consensus motif	ψKXE/D	Rodriguez et al. 2001 [75]
Inverted Consensus motif	E/DXKψ	Matic et al. 2010 [76]
Hydrophobic cluster motif	(ψ)_n_KXE	Matic et al. 2010 [76]
PDSM	ψKXEXXS-℗	Hietakangas et al. 2005 [77]
NDSM	ψKXEXX(E)_n_	Yang et al. 2006 [78]

Ψ = bulky hydrophobic residues; K = the lysine modified; E/D = acidic residues; X = any residue; n = two or more; S-℗ = phosphorylated serine.

## Data Availability

Not applicable.

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
