# Peer review of "Role of SUMOylation in Neurodegenerative Diseases"

_cells, 2022, doi:10.3390/cells11213395_

Round 1
Reviewer 1 Report
The manuscript entitled "Role of SUMOylation in neuronal disease" is an interesting-to-read review with a major focus on the role of SUMO-based protein modification in neurodegenerative disease, such AD and PD, as well as diabetic peripheral neuropathy. The reviews covers both molecular mechanisms of SUMOylation as well as key and recent reports on this post translational modification of proteins in neuronal diseases, in particular during ageing. Therefore, publication is highly encouraged by this referee. The only minor point is that the authors shall check their English language, which can be improved in some passages of this manuscript.
Author Response
Suggestion: to improve English text
Response: The authors are highly grateful to the reviewer, and we have redrafted the manuscript to improve the language.
Reviewer 2 Report
Neurodegenerative diseases are a major problem in an ageing society. Many efforts are being devoted to understanding the underlying molecular mechanisms at the molecular level. Comprehensive and well-written reviews on specific aspects of neurodegeneration are valuable, especially when they add new aspects. However the text “Role of SUMOylation in Neuronal disease” is based in a big part on previous reviews, and only one publication from 2022 is cited. Please add some new publications in to the text, for example:
doi: 10.1016/j.neulet.2022.136771.
doi: 10.1016/j.chembiol.2020.12.010.
doi: 10.1016/j.ibneur.2022.01.003
Moreover, extensive English proofreading is required throughout the manuscript. In several places, the true meaning of the text is missed due to language problems.
Minor corrections, probably some of them will be eliminated during English correction.
In the title, the use of the singular form Neuronal disease instead of neurodegenerative diseases is misleading.
In key words it should be Alzheimer’s, Parkinson’s and Huntington’s disease.
Line 28 What are the specific tools?
Line 46 amino acid residues
Line 74 The PML abbreviation should be introduced
Line 88 Proteins always have localization, it is impossible to facilitate localization. It could be that modification results in the changes in protein localization.
Line 96 repair of what?
Line 101 What are the individual pathways?
Line 104 e-amino should be ε-amino
Line 107 There is no information that translation results in production of SUMO precursors.
Line 116 E2 ligase?
Lines 115-117 The sentence has to many different pieces of information.
Lines 119-120 What is SUMOylation adaptability?
Line 127 It is impossible to transfer motif.
Line 142 In table there are 4 groups of Ligases in a text that there are 3.
Line 152 SUMO motif transfer – What is SUMO motif? Motif cannot be transferred.
Lines 155-156 what is consensus sequence? It is not clear -SIM sequence, but the consensus is not given?
Lines 175-177 I don’t understand this sequence what is magnified?
Table 2 . Left only title above the table, the rest should be below.
Lines 186-187 This is repetition of the sentence in lines 51-52.
Line 196 AD is associated with […] and with loss of neuropil threads – but neuropil threads are rather considered as a toxic elements the morphological hallmarks of AD.
Line 218 There is no information about APP cleavage by β secretase, it is only general information that APP is precursor. Later on BACE1 name is used. Authors should use the same nomenclature consistently throughout the text.
Line 263 What is imprudent membrane potential?
Line 276 Mutations are in gene not in a protein, on a level of proteins the amino acid residues substitutions are observed
Line 277 autosomal dominance – do Authors mean autosomal dominant inheritance?
Line 290 What is a link between SNARES and microtubule assembly? Additionally, SNARE complexes should be introduced.
Line 297 First residue 87 next 129
Line 313 Neurons are not the reason of oxidative stress etc., but lack of PARK2.
Line 318 Substitutions in DJ-1
Line 329 glutamines are not present in a gene
line 332 the gene does not play a role in axon trafficking
Line 334 What is bidirectional switching? It is not clear from the text.
Line 338 “Htt with polyQ repeats mitigate the HD disease pathology mainly in 338 spiny neurons to the striatum and pyramidal neurons that project to the striatum.” Wild-type Htt also has polyQ, I don’t understand sense of the sentence.
Line 351 The exon cannot be SUMOylated only protein encoded by the exon.
Line 361 What is mHtt?
Line 367 Protein upregulation – increased level of protein, its activity or both – specify.
Line 368 The summary of the part 5 is strange, it was not only about Htt and mitochondria.
Lines 390 and 397 (GAPDH) introduced in line 390, so use GAPDH in line 397
Line 402 “SUMOylated enzyme involved in” – which enzyme?
Lines 405-407 The sentence is not easy to understand
Line 412 It is impossible to delete cascade
Lines 416-419 Is the list of all channels required , as only introduction of TRPV1 is necessary. Please simplify the text, by removing unnecessary names.
Lines 440 and 441 The abbreviations IkB and NFkB to be used here have to be introduced earlier.
Author Response
Reviewer 2 comments
Neurodegenerative diseases are a major problem in an ageing society. Many efforts are being devoted to understanding the underlying molecular mechanisms at the molecular level. Comprehensive and well-written reviews on specific aspects of neurodegeneration are valuable, especially when they add new aspects. However the text “Role of SUMOylation in Neuronal disease” is based in a big part on previous reviews, and only one publication from 2022 is cited. Please add some new publications in to the text, for example:
doi: 10.1016/j.neulet.2022.136771.
doi: 10.1016/j.chembiol.2020.12.010.
doi: 10.1016/j.ibneur.2022.01.003
Response: Authors are grateful to the reviewer. We added more references to enrich the content of the revised manuscript.
Moreover, extensive English proofreading is required throughout the manuscript. In several places, the true meaning of the text is missed due to language problems.
Response: We have redrafted the text to increase clarity.
Minor corrections, probably some of them will be eliminated during English correction.
Response: We have redrafted the manuscript to improve the language.
In the title, the use of the singular form Neuronal disease instead of neurodegenerative diseases is misleading.
Response: The title is changes to “Role of SUMOylation in neurodegenerative diseases” as per reviewer advice.
In key words it should be Alzheimer’s, Parkinson’s and Huntington’s disease.
Response: The key words are changed in the revised manuscript.
Line 28 What are the specific tools?
Response: The specific tools refers to diagnostic tools to identify NDDs. The text is accordingly modified.
Line 46 amino acid residues
Response: text changed to residues
Line 74 The PML abbreviation should be introduced
Response: PML is now explained in the text.
Line 88 Proteins always have localization, it is impossible to facilitate localization. It could be that modification results in the changes in protein localization.
Response: The text is modified accordingly.
Line 96 repair of what?
Response: DNA repair, the text is modified to bring clarity.
Line 101 What are the individual pathways?
Response: Text is modified to explain the impact of SUMOylation on signalling pathways.
Line 104 e-amino should be ε-amino
Response: Changed in the text.
Line 107 There is no information that translation results in production of SUMO precursors.
Response: The information is added to explain the expression and processing of SUMO precursors.
Line 116 E2 ligase?
Response: Change to E2 conjugated.
Lines 115-117 The sentence has to many different pieces of information.
Response: Text is rewritten to bring clarity.
Lines 119-120 What is SUMOylation adaptability?
Response: Text is modified to bring clarity.
Line 127 It is impossible to transfer motif.
Response: Authors agree with reviewer and change the text in entire manuscript accordingly.
Line 142 In table there are 4 groups of Ligases in a text that there are 3.
Response: The explanation in the text is modify to synchronized with the table summarizing the types of E2 ligases.
Line 152 SUMO motif transfer – What is SUMO motif? Motif cannot be transferred.
Response: Text changed accordingly as mention before.
Lines 155-156 what is consensus sequence? It is not clear -SIM sequence, but the consensus is not given?
Response: The text explaining SIM consensus sequence is added to the manuscript.
Lines 175-177 I don’t understand this sequence what is magnified?
Response: The explanation is added to bring clarity.
Table 2 . Left only title above the table, the rest should be below.
Response: Changed as per reviewer advice.
Lines 186-187 This is repetition of the sentence in lines 51-52.
Response: Sentence deleted.
Line 196 AD is associated with […] and with loss of neuropil threads – but neuropil threads are rather considered as a toxic elements the morphological hallmarks of AD.
Response: We have rephrased this part of the text and added more information.
Line 218 There is no information about APP cleavage by β secretase, it is only general information that APP is precursor. Later on BACE1 name is used. Authors should use the same nomenclature consistently throughout the text.
Response: The text is added to explain APP processing by β secretase. The manuscript is proofread to use the same nomenclature for β secretase.
Line 263 What is imprudent membrane potential?
Response: Text is rephrased to explain the impact of α-synuclein on membrane protential.
Line 276 Mutations are in gene not in a protein, on a level of proteins the amino acid residues substitutions are observed
Response: Manuscript is proof read to adapt to standard writing standard for gene and protein and accordingly changes are made in the entire manuscript.
Line 277 autosomal dominance – do Authors mean autosomal dominant inheritance?
Response: Additional text and explanation added to the manuscript to bring clarity.
Line 290 What is a link between SNARES and microtubule assembly? Additionally, SNARE complexes should be introduced.
Response: Text is rephrased, and additional information is added to describe SNARE complex and synuclein-induced modulation of SNARE and microtubules.
Line 297 First residue 87 next 129
Response: Changed accordingly.
Line 313 Neurons are not the reason of oxidative stress etc., but lack of PARK2.
Response: Text is rephrased to bring clarity.
Line 318 Substitutions in DJ-1
Response: DJ-1 is explained in the revised manuscript.
Line 329 glutamines are not present in a gene
Response: Text is rephrased to avoid any discrepancy.
line 332 the gene does not play a role in axon trafficking
Response: Text is rephrased for clarity.
Line 334 What is bidirectional switching? It is not clear from the text.
Response: New text is added to explain HTT-mediated bidirectional axonal transport.
Line 338 “Htt with polyQ repeats mitigate the HD disease pathology mainly in 338 spiny neurons to the striatum and pyramidal neurons that project to the striatum.” Wild-type Htt also has polyQ, I don’t understand sense of the sentence.
Response: The text is rephrased and new explanation is added to bring clarity.
Line 351 The exon cannot be SUMOylated only protein encoded by the exon.
Response: The text is rewritten to avoid and confusion.
Line 361 What is mHtt?
Response: More information is added to explain mHTT.
Line 367 Protein upregulation – increased level of protein, its activity or both – specify.
Response: Text added to explain alteration in protein levels as well as in protein activity.
Line 368 The summary of the part 5 is strange, it was not only about Htt and mitochondria.
Response: New summary is added to the section.
Lines 390 and 397 (GAPDH) introduced in line 390, so use GAPDH in line 397
Response: modified as per reviewer’s advice
Line 402 “SUMOylated enzyme involved in” – which enzyme?
Response: Explanation added to the manuscript.
Lines 405-407 The sentence is not easy to understand
Response: Rewritten the sentence to bring clarity.
Line 412 It is impossible to delete cascade
Response: Text added and modified to avoid any confusion.
Lines 416-419 Is the list of all channels required , as only introduction of TRPV1 is necessary. Please simplify the text, by removing unnecessary names.
Response: Authors believe it is important to mention the list of identified channels which are SUMOylation target. Despite some many channel can be modified/regulated by SUMOylation only TRPV1 is studies in details in DPN.
Lines 440 and 441 The abbreviations IkB and NFkB to be used here have to be introduced earlier.
Response: The text added to explain IkB and NFkB.
Round 2
Reviewer 2 Report
See attached file.

Author Response
Response to reviewer’s comments
I have mainly minor comments on wording, word order and additionally a few editorial comments. These are listed below.
Part 2- consider changes in the order of information. I find it logical to present SUMO first, then the SUMO conjugation system (lines 100-164), then provide information on the role of SUMOylation in physiology (lines 76-99), and finally information on the disease state. If you decide not to change the order, then remove 'SENP protease-mediated' from lines 80-81, as SENP is introduced later.
Response: We would like to keep the current format and as per reviewer advice we have deleted the text 'SENP protease-mediated' from lines 80-81.
lines 22 and 30 the same information “financial burden on society”
Response: The Sentence is rephrased to avoid any repetition.
line 39 presence/appearance of non-functional chaperones?
Response: The text “dysfunctional molecular chaperones” implies that PTM may cause alteration in chaperones function which could contribute to irregularities in cellular function.
line 42 It will be better to use: attaching of different…
Response: Changed as per reviewer advice.
line 66 regulateds
Response: Changed as per reviewer advice.
It will be better to write:”… studies have shown…”
Response: Changed as per reviewer advice.
Line 68 expressed in tissues
Response: Changed as per reviewer advice.
line 71 The sentence "Promyelocytic..." is unfinished.
Response: The sentence is rephrased.
line 76 switching between
Response: Changed as per reviewer advice.
lines 76 and 83 repetition of information
Response: Text rephrased to avoid and repetition.
line 85 substrate modified protein
Response: Changed as per reviewer advice.
Line 86 This is similarity in role of SUMO attachment to other PTM not alternative, at least according to text.
Response: SUMOylation may or may not counteract the function of other PTMs. Here in line 82 (previously line 86) the impact of SUMOylation of occurrence of other PTM is mentioned. It is shown in several studies the SUMOylation of substrate trigger other PTM to occur.
Line 87 Remove word: substrate
Response: Changed as per reviewer advice.
Line 88 Desired subcellular compartment
Response: We would prefer to use word “targeted subcellular compartment” rather than desired subcellular compartment.
Line 95 Not clear growth of what
Response: Rephrased the sentence.
Line 104 Produced are SUMO forms
Response: We prefer to use the word “expressed/expressing” of protein from genes as compared to “produced/producing” in Invivo situation as later text confused reader with Invitro protein synthesis methods
Line 110 Diglycine motif of SUMO
Response: Changed as per reviewer advice.
Line 113 The UBE2I is a name of the gene and UBC9 of protein, so UBE2I is not an enzyme – correct.
Response: UBE2I is name of gene and its encoded protein (UBE2I) for human analogues, whereas UBC9 is name of gene and its encoded protein (UBC9) in rodents and Yeast
Line115 Duplication of the sentence –remove
Response: Changed as per reviewer advice.
Line 131 motif of the substrate protein
Response: Changed as per reviewer advice.
Line 139 In figure there is DeSUMOylation so in caption should be the same.
Response: Changed as per reviewer advice.
Line 149 have has
Response: Changed as per reviewer advice.
line 170 or aspartic acid residues, respectively
Response: Changed as per reviewer advice.
line 173 What is the P in the end of PDSM motif?
Response: P is proline residue in the PDSM motif (ψKXEXX(pS)P), description added to the text.
line 179 Table 2 collectively presents the different
Response: Changed as per reviewer advice.
line 186 modification of the ion channels properties
Response: Changed as per reviewer advice.
line 214 disparities between?
Response: Rephrased the text.
Line 219 LTP? This is abbreviation used here for a first time – give full name.
Response: Changed as per reviewer advice.
Lines 220-221 UBC9 expression – it is unclear whether the word expression refers to the expression of the UBE2I gene encoding the UBC9 protein, i.e. an increase in the amount of mRNA, or whether it is an increase in the level of the UBC9 protein. So, UBE2I expression if gene or UBC9 level if protein
Response: Rephrased the text for clarity.
Line 223 facilitate in the synaptic
Response: Changed as per reviewer advice.
Line 229 Expressing genes encoding enzymes of SUMOylation pathway
Response: Changed as per reviewer advice.
Line 230 of APP-cleaving protease β-secretase encoding gene
Response: Changed as per reviewer advice.
Line 232 Is pathology regulated? Maybe better SUMOylation-mediated pathological changes in AD.
Response: Changed as per reviewer advice.
Line 247 There is a logical gap between sentences.
Response: Rephrased the text.
Line 256 in which SUMOylation plays a role in AD
Response: Changed as per reviewer advice.
Line 324 In contrast instead of contradictory
Response: Changed as per reviewer advice.
Line 327 Substitutions of lysine…
Response: Changed as per reviewer advice.
Line 330 Producing His-tagged
Response: We prefer to use the word “expressed/expressing” of protein from genes as compared to “produced/producing” in vivo situation as later text confused reader with invitro protein synthesis methods
Line 332 10% of SUMOylated
Response: Changed as per reviewer advice.
Line 331 – maintain the same way of naming Lysine 102
Response: Changed as per reviewer advice.
Line 337 production of
Response: Same as in comment line 330.
Line 338 Drosophila
Response: Changed as per reviewer advice.
Line 339 with the parkin protein (encoded by the PARK2 gene).
Response: Changed as per reviewer advice.
Line 340 not only one mutation Mutations in PARK2 are the cause of autosomal…
Response: Changed as per reviewer advice.
Line 341 …lysosomal degradation pathways.
Response: Changed as per reviewer advice.
Line 345 activity and decreases its levels. I am confused because there is contradictory information in line 345 and line 348.
Response: Text is rewritten to bring clarity for reader.
Line 353 Leucine 166 Proline substitution
Response: Changed as per reviewer advice.
Line 357 proteins involved in mitochondrial
Response: Changed as per reviewer advice.
line 368 encoding
Response: Changed as per reviewer advice.
line 370 the presence of more than 36 CAG triplet repeats encoding polyQ tract in HTT protein
Response: Changed as per reviewer advice.
line 374 – not a sentence Antero- and retro grade axon transport [137,138].
Response: Rephrased the sentence.
Lines 385-387 Repeated info from line 366
Response: Rephrased the sentence.
Line 397 In contrast, SUMOylation of protein fragment encoded by exon 1 of the HTT gene
Response: Changed as per reviewer advice.
Lines 400-401 the domain was not introduced interaction with lipids mediated by the Nt17 lipid binding domain to prevent aggregation
Response: Changed as per reviewer advice.
line 403 PIAS1, through interaction with HTT, is also part of the transcription-coupled repair complex (TCR) and polynucleotide kinase 3′-phosphatase (PNKP).
Response: Changed as per reviewer advice.
Line 416 exhibited resulted in a
Response: Changed as per reviewer advice.
line 457 a few studies attempted
Response: Changed as per reviewer advice.
lines 459-460 critical enzymes in glycolysis and the Krebs cycle as targets for SUMOylation
Response: Changed as per reviewer advice.
line 463 are SUMOylation targets
Response: Changed as per reviewer advice.
line 464 analysis showed
Response: Changed as per reviewer advice.
line467 depicted
Response: Changed as per reviewer advice.
line 468 stabilizes its expression – stabilization of expression or stabilization of protein? If the second than it should be and stabilizes it.
Response: Rephrase the sentence.
line 469 It This results […] STAT3 (Signal transducer and activator of transcription 3)
Response: Changed as per reviewer advice.
line 474 UBC9 is a name of protein not a gene
Response: UBC9 is name of gene as well as the encoding protein. Please see the explanation in previous comment for line 113.
Lines 495 and 519 What DRG is?
Response: Introduced in line 458.
Line 504 and 505 The abbreviations NF-kB and IkB have already been introduced (lines 65-66), so just leave the abbreviations here.
Response: Changed as per reviewer advice.
Line 508 It induces formation of disulfide bond
Response: Changed as per reviewer advice.
Line 509 To reduced levels
Response: Changed as per reviewer advice.
Lines 511-514 I don’t understand the sentence.
Response: Rephrased he sentence for clarity.
Figures There are two Figures 1
Response: We are thankful to reviewer for mentioning this mistake. We have corrected all figure numbers accordingly.
In Figures on the right site should be Alzheimer’s disease, Parkinson’s disease, Huntington’s disease.
Response: Changed as per reviewer advice.
Figure (1) in lines 257-259 You can’t have site directed mutagenesis in healthy organism. This is an artificial situation.
Response: The figures are modified to avoid any kind of confusion.
In the AD conditions there is a confusing arrow from a degraded secretase with the signature αcleavage of APP. There is no explanation that this is by another secretase.
Response: The figures are modified to avoid any kind of confusion.
Figure 2 Dopamine and dopamine receptor Lewy bodies
Response: Changed as per reviewer advice.
I don’t understand. Does at the same time sumoylation of α-synuclein result in decreased solubility (so it should result in aggregation) and decreased fibres formation? With reference to the text of line 227, perhaps there should be an increase in solubility
Response: The figures are modified to avoid any kind of confusion.